# Computational Characterization of the Dish-In-A-Dish, A High Yield Culture Platform for Endothelial Shear Stress Studies on the Orbital Shaker

**DOI:** 10.3390/mi11060552

**Published:** 2020-05-29

**Authors:** Rob Driessen, Feihu Zhao, Sandra Hofmann, Carlijn Bouten, Cecilia Sahlgren, Oscar Stassen

**Affiliations:** 1Department of Biomedical Engineering, Eindhoven University of Technology, 5600 MB Eindhoven, The Netherlands; r.c.h.driessen@tue.nl (R.D.); feihu.zhao@swansea.ac.uk (F.Z.); s.hofmann@tue.nl (S.H.); c.v.c.bouten@tue.nl (C.B.); c.m.sahlgren@tue.nl (C.S.); 2Institute for Complex Molecular Systems, Eindhoven University of Technology, 5600 MB Eindhoven, The Netherlands; 3Zienkiewicz Centre for Computational Engineering, College of Engineering, Swansea University, Swansea SA1 8EN, UK; 4Faculty of Science and Engineering, Biosciences, Åbo Akademi University, 20500 Turku, Finland; 5Turku Bioscience Centre, Åbo Akademi University and University of Turku, 20520 Turku, Finland

**Keywords:** shear stress, computational fluid dynamics, endothelial cells, orbital shaker, flow

## Abstract

Endothelial cells sense and respond to shear stress. Different in vitro model systems have been used to study the cellular responses to shear stress, but these platforms do not allow studies on high numbers of cells under uniform and controllable shear stress. The annular dish, or dish-in-a-dish (DiaD), on the orbital shaker has been proposed as an accessible system to overcome these challenges. However, the influence of the DiaD design and the experimental parameters on the shear stress patterns is not known. In this study, we characterize different designs and experimental parameters (orbit size, speed and fluid height) using computational fluid dynamics. We optimize the DiaD for an atheroprotective flow, combining high shear stress levels with a low oscillatory shear index (OSI). We find that orbit size determines the DiaD design and parameters. The shear stress levels increase with increasing rotational speed and fluid height. Based on our optimization, we experimentally compare the 134/56 DiaD with regular dishes for cellular alignment and *KLF2*, *eNOS*, *CDH2* and *MCP1* expression. The calculated OSI has a strong impact on alignment and gene expression, emphasizing the importance of characterizing shear profiles in orbital setups.

## 1. Introduction

Endothelial cells line the vasculature and are sensitive to fluid shear stress induced by the blood flow. The cells sense shear stress and convert mechanical stress to biological signals to control vascular tone and morphology [1,2]. The shear stress acting on the endothelial cells is directly linked to vascular homeostasis and disease [3,4]. Oscillatory and low laminar (<0.2 Pa) shear stress patterns are atheroprone. Cells subjected to this type of shear stress align poorly and show an increased expression of proinflammatory markers such as monocyte chemoattractant protein 1 (*MCP1*), a chemokine involved in monocyte attraction, and cadherin-2 (*CDH2*), a mesenchymal marker [5,6,7,8,9]. High laminar flow profiles (shear stresses > 1 Pa) are considered atheroprotective [7,10]. Cells that are exposed to an atheroprotective flow align in the direction of the flow, form a tight vascular barrier and express anti-inflammatory genes [11]. Typical anti-inflammatory markers are Krüppel-like Factor 2 (*KLF2*) and endothelial nitric oxide synthase (*eNOS*), both classical shear stress response markers found in healthy endothelium [12,13,14]. 

Different culture systems are used to study endothelial cell behavior under shear stress. The most commonly used system is the parallel plate setup [15,16,17]. In this system, medium is pumped through a rectangular or circular channel into which cells are seeded. The shear stress acting on the cells is determined by the flow speed and channel dimensions. However, the number of cells that can be exposed to shear stress simultaneously is limited in this set-up. Another frequently used approach is the cone and plate apparatus, where cells are seeded in a medium-filled dish and a cone is placed in the medium [15]. By rotating the cone, a fluid flow is created exerting a shear stress on the cells. The shear stress can be calculated from the viscosity of the medium, the cone angle and the rotational speed of the cone. In this system only a single dish of cells can be exposed to shear stress at once, again limiting the number of cells at study. As an alternative for molecular studies that require high number of cells (e.g., proteomics or chromatin conformation capture), an orbital shaker can be used [15]. By placing a culture dish or plate on a shaker, a frictional drag force is created on the cells. As a result, a non-uniform shear stress field is created that depends on the angular velocity and the radius of the shaker. Although this approach increases the experimental throughput significantly, the distribution of the shear stress is inhomogeneous and includes both atheroprone and atheroprotective regions [18,19]. These conflicting mechanical cues lead to conflicting biological responses. Consequently, variance is introduced in the system and false conclusions may be drawn from experiments using an orbital shaker. 

To overcome this issue, while retaining high throughput, an annular dish culture setup has been proposed [20,21,22]. This is an easy-to-implement platform in which a small standard culture dish is mounted into a larger standard dish. In this dish-in-a-dish (DiaD) setup, fluid travels around in the donut-like shaped culture vessel, promoting unidirectional fluid flow at all time. It is important to know the shear stress that the cells are exposed to since this governs atheroprone and -protective phenotypes. A commonly used approach to estimate the maximum shear stress values is the extended solution of Stokes’ second problem [23,24]. The resulting shear stress values are based on the orbit size of the orbital shaker and the angular velocity. This analytical approach assumes unperturbed fluid in the far field, neglecting the side walls and initial fluid height [19,25,26]. Computational fluid dynamics (CFD) has been shown to be a more precise method to characterize flow and shear stress patterns in dishes on the orbital shaker [18,27,28,29]. Simulations by these groups have shown that the shear stress distribution is non-uniform, confirming that the analytical approximation cannot be used for the characterization of shear stress profiles on the orbital shaker. Since the previously available characterization of the DiaD shear distribution was based on the analytical approach [20,21], a more accurate characterization of the shear stress distribution in the DiaD is needed. In this work, we characterize the influence of orbit size, orbital speed and fluid height on the shear stress distribution across the DiaD using CFD. Based on these parameters, different dishes are recommended for different orbit sizes. These can be optimized for either maximal shear, homogeneity of shear or cell numbers. Finally, the endothelial shear stress response in the DiaD was validated by cell orientation and gene expression analysis.

## 2. Materials and Methods 

### 2.1. Computational Approach

A computational fluid dynamics (CFD) model was developed in ANSYS-CFX (ANSYS Inc., Canonsburg, PA, USA) to model the shear stress in (annular) dishes on the orbital shaker. The influence of orbit diameter, rotational speed and fluid levels was evaluated. Commercially available shakers typically have an orbital diameter (d_ob_) of 10, 19 or 25 mm. The orbital speeds were tested at 100, 125, 150 and 200 rpm and the influence of initial fluid levels was examined for 2, 3 and 4 mm of fluid. Unless otherwise stated, an orbital speed of 200 rpm and a medium height of 3 mm was used. 

The medium was modelled as an incompressible and Newtonian fluid with a dynamic viscosity (*µ*) of 0.7 mPa·s at 37 °C [30]. To determine whether the flow was laminar or turbulent, we first estimated the Reynolds number (Re) using
(1)$$Re=\frac{\rho \;\omega \;{d_{ob}}^{2}}{\mu }$$
where *ρ* is the medium density (*ρ* = 1000 kg/m^3^) and ω is the orbital speed (ω = 20.94 rad/s for 200 rpm) [31]. Afterwards, the exact *Re* number was calculated from the CFD pre-computation. Based on this first approximation is the 10-mm orbit in the transition zone between laminar and turbulent flow with a *Re* of approximately 3000. The other orbit diameters resulted in an approximate *Re* of 11000 and 18500 (19 and 25 mm, respectively), indicating turbulent flow. Therefore, we first simulated both laminar and turbulent flow. For the turbulent flow, a κ-epsilon turbulence model was incorporated into the Navier–Stokes equation [29]. Both methods yielded high *Re* numbers in the turbulence range. Therefore, the turbulence model was used in all simulations.

The pressure drop across the medium–air interface was expressed by the radius and the surface tension was calculated as
(2)$$p_{M}-p_{A}=\gamma \left(\frac{1}{r_{M}}+\frac{1}{r_{A}}\right)$$
where *p_M_* and *p_A_* are the pressures measured by the surface curvature in the normal direction to the interface in liquid (medium) and gas (air) phases, respectively; *γ* is the surface tension of the medium (*γ* = 0.072 N/m) [29] *r_M_* and *r_A_* represent the radius measured by the surface curvature in the normal direction to the interface in liquid (medium) and gas (air) phases, respectively.

The volume of fluid (VOF) technique, which has been introduced in previous studies, was used for tracking the medium–air interface during shaking [27,28,32]. In each computational finite volume at the interface, the continuity and momentum equations were solved based on the modified definition of the fluid properties *P* [28,29]:(3)$$\{\begin{array}{c} P=\alpha_{M}\cdot P_{M}+\alpha_{A}\cdot P_{A}\\ \;\\ \alpha_{M}+\alpha_{A}=1 \end{array}$$
where *P_M_* and *P_A_* are the properties of the medium and air (i.e., density and dynamic viscosity), *α_M_* and *α_A_* are the volume fraction of medium and air, respectively.

The dish wall was rigid with a no-slip boundary condition. The wall shear stress *τ* on the substrate of the dish was calculated by
(4)$$\tau ={\mu \cdot \frac{\partial u}{\partial z}|}_{z=0}$$
where *u* is the fluid velocity parallel to the substrate and *z* is the distance to the substrate in coordinate O’ (Figure 1). 

The fluid domain was meshed by hexahedral elements, with D the outer diameter and d the inner diameter of the annular dish (Figure 1A). After mesh sensitivity analysis, it was found that the mesh element size of 1500 µm was sufficient for eliminating the mesh-dependent effect (Appendix A). When modeling dishes without an inner dish d was set to 0. As dish dimensions standard culture dishes were used. In contrast to what the product name suggests (35, 60, 100 or 150 mm), the actual diameters of these dishes (35, 56, 89 and 134 mm, respectively) were used for the mesh. Transient analysis was used in the simulation with a time step of 0.01 s for a whole-time length of 4.0 s. Finally, the CFD model was solved by a finite volume method (FVM) using ANSYS CFX under the convergence criteria of the root mean square residual of the mass and momentum < 10^−^^4^. 

For each dish, the radial shear stress distribution was normalized between d and D. To analyze the shear stress profiles over one period in the annular dish designs, all elements were grouped in n groups based on their radial position (r_n_). For the regular culture dishes, 20 different time points in one period were analyzed since the number of elements at d close to zero is limited. From these temporal data, the maximum (τ_max,r_) shear stress was determined for each radial position (Figure 1B). Next, the oscillatory shear index (OSI), a measurement method for the degree of variation in the shear direction, was identified for each radial position using: (5)$$OSI=1-\frac{\left|\int_{0}^{T}WSSdt\right|}{\int_{0}^{T}\left|WSS\right|dt}$$
where *T* is one orbital period. The main shear direction was defined by the direction of the sum of all shear stress vectors at each radial position. For this, all vectors were first rotated based on their radial coordinates.

### 2.2. Experimental Validation

#### 2.2.1. DiaD Fabrication

To make the dish-in-a-dish system, a small culture dish was mounted in a larger dish using polydimethylsiloxane (PDMS, Sylgard 184, Dow Corning, Midland, MI, USA) with standard culture dishes 60 and 150 mm in diameter. The rim of the inner dish was dipped in PDMS pre-polymer (10:1 *w*/*w* ratio of PDMS to curing agent) and placed in the center of the large culture dish. A small hole was drilled in the inner dish to allow expanding air to escape during curing at higher temperature. The PDMS was cured overnight at 65 °C. Afterwards, the annular dish was stored until further use. 

#### 2.2.2. Cell Culture

Pooled human umbilical vein endothelial cells (HUVEC, Lonza, Basel, Switzerland) were cultured in endothelial growth medium, supplemented with growth factors (EGM-2 medium, Promocell, Heidelberg, Germany). The cells were subcultured upon 80% confluence and the medium was changed every 2–3 days. The cells were used until passage 5. All cell cultures and experiments were performed at 37 °C and 5% CO_2_ in a humidified incubator. 

#### 2.2.3. Shear Stress Experiments

The annular dish was sterilized with 70% ethanol for 30 min under UV light. Afterwards, the dish was rinsed twice with phosphate-buffered saline (PBS, Sigma Aldrich, St. Louis, MO, USA). After sterilization, the annular dish was coated with collagen IV in PBS (0.85 μg/cm^2^) for 1 h at 37 °C and afterwards washed with PBS once. HUVEC were seeded at 25,000 cells/cm^2^ density in the annular dish and grown to a monolayer. Upon confluence, the medium was replaced with 35 mL (corresponding with 3 mm) medium and the annular dishes were fixed on a horizontal orbital shaker (Yellowline OS 10 Basic, IKA GmbH, Staufen, Germany) in the incubator (Appendix A). Shear stress was generated by shaking the annular dish at 200 rpm for 24 h. The static controls were cultured in the same incubator. 

After 24 h, the mechanically loaded cells and their respective static controls were imaged using an EVOS system (Thermo-Fisher, Waltham, MA, USA). The orientation of the cells was analyzed using the ImageJ directionality tool. After imaging, the cells were washed twice with cold PBS and removed from the dish by scraping. The cells were collected in PBS and spun down at 150 g for 5 min. The pellet was lysed in RLT buffer (RNeasy kit, Qiagen, Hilden, Germany) and stored at −80 °C for gene expression analysis. All experiments were performed in three experimental replicates. 

#### 2.2.4. Gene Expression Analysis

RNA was isolated using the Qiagen RNeasy kit according to the manufacturer’s instructions and as previously described [33]. For each sample 140 ng of RNA was synthesized to cDNA by using the SuperScript VILO cDNA synthesis kit (Invitrogen). Reference gene stability was tested for six different genes, analyzed by GeNorm [34], and glyceraldehyde 3-phosphate dehydrogenase (*GAPDH*) was selected as the most stable reference gene. The PCR protocol consisted of a 3 min 95 °C incubation, followed by 40 cycles of 20 s at 95 °C, 20 s at 60 °C and 30 s at 72 °C. Data were analyzed using the ΔΔCt method. The primer sequences can be found in Appendix A. The *GAPDH* primer was purchased from PrimerDesign (Southampton, UK). 

#### 2.2.5. Statistical Analysis

All experiments were performed in triplicate as three independent experiments. Data are presented as the mean +/- standard deviation. Statistical significance between the static and flow conditions was evaluated with Student’s t-test using GraphPad Prism 6 software, *p* < 0.05 was considered significant.

## 3. Results

The flow characteristics that define atheroprone and atheroprotective regimes are complex, but it has been established that low τ_max_ and OSI are atheroprone and correlate with atherogenesis [19,35]. We aimed for dishes with a high τ_max_ and a low OSI, while maximizing the cell culture area in the dish. For this optimization process, we first investigated whether the DiaD design resulted in an improved flow pattern over standard dishes, and how τ_max_ and OSI were affected by the different geometries. Next, we tested the different orbits available from commercial suppliers, as these will be fixed depending on the equipment available in the research environment. We further optimized, for the influence of the orbital speed and volume, parameters that effected the shear stress patterns.

### 3.1. Optimization of Dish Design

First, we characterized the conventional culture dishes. The shear stress profile in a standard six-well culture dish and in a 15-cm culture dish, further referred to by their actual dimensions of 35 and 134 mm, were modeled on an orbital shaker (10-mm orbit) at 200 rpm. The shear stress levels in the center of both dishes were close to zero and displayed a high OSI (Figure 2), corresponding to atheroprone flow patterns. At the edge of the 35-mm dish, the shear profile obtained an atheroprotective characteristic with a maximum shear stress of 0.75 Pa and OSI values close to zero. The larger 134-mm dish, on the other hand, displayed lower shear levels throughout the dish with a τ_max_ of 0.41 Pa. The OSI levels initially dropped to 0.14 but then increased again to stabilize around 0.45–0.50. These data showed that upscaling from a 35-mm dish to a larger 134-mm dish affects shear stress and results in a relative increase in the area with an atheroprone flow pattern.

We next characterized the different annular designs of the DiaD that can be obtained by adapting conventional culture dishes. The resulting annular dishes were divided into three groups based on the width of the culture channel vs. the dish size: narrow, intermediate and wide designs (Figure 3). We modeled the shear stress in these designs on a 10-mm orbit at 200 rpm with an initial fluid height of 3 mm. In the narrow and intermediate dish designs the highest shear stress values were found at the inner dish. In the narrow dish, the largest τ_max_ was found in the larger dish, 0.40 Pa for the 134/89 (outer/inner diameter) and 0.28 Pa for the 89/56 dish. In these dishes, the τ_max_ slowly decreased towards the outer edge. This coincided with a peak in the OSI halfway the culture area. The height of the OSI peak is dependent on the size of the annular dish, the larger dish reached a higher OSI (0.37 for the 134/89 dish and 0.19 for the 89/56 dish). The intermediate sized dishes started at a τ_max_ of 0.85 Pa, but this value decreased when moving to 0.15–0.25 Pa at the outer edge. The OSI behavior is similar to the narrow dishes, the bigger dish displayed a higher maximum OSI (0.35 over 0.18 respectively). The wide dish design displayed a peak τ_max_ (0.53 Pa) in the center of the culture area with an OSI of 0.33, like the other designs with a 134-mm outer dish. For further analysis, we tested the dishes with a 134-mm outer dish, since these offer the largest culture area and highest cell numbers. Since the intermediate sized annular dish could obtain large τ_max_ values and the narrow dishes resulted in low variations in τ_max_, these two designs were further characterized.

### 3.2. Optimization of Experimental Parameters

The most commonly used orbital shakers have orbit sizes of 10, 19 or 25 mm, respectively. Therefore, we characterized the shear stress distribution in the annular dish on these orbits. In general, the τ_max_ increased with increasing orbit size (Figure 4A). The variation in τ_max_ is lowest on the smaller 10-mm orbit size. On the 19-mm orbit, the variation in τ_max_ is lowest in the 134/89 dish. The OSI is also lower in this dish on the 19-mm orbit (Figure 4B). The 134/56 dish on a 19-mm orbit resulted in a high variation in τ_max_ throughout the dish, with low shear stress values close to zero at the inner dish and up to 2 Pa in the center of the culture area. A similar distribution in τ_max_ was found for both dish designs on the 25-mm orbit. Because of the high variation in τ_max_, the dishes on the 25-mm orbit were not further characterized. The 134/89 design on the 19-mm orbit displayed lower variation in τ_max_ and comparable values with the results on the 10-mm orbital shaker. Taken together, the 134/56 dish is the optimal dish on the 10-mm orbit, based on τ_max_ levels. On the 19-mm orbital shaker, the 134/89 dish is preferred over the 134/56 dish, based on the τ_max_ distribution and OSI. For that reason, these two dishes were further characterized on a 10-mm and 19-mm orbit, respectively.

An important parameter for dishes on the orbital shaker is the orbital speed. In this work, we tested the effect of orbital speed on the shear stress distributions by modeling the orbital speed from 100 to 200 rpm (Figure 5A). For both dishes, the τ_max_ increased with increasing orbital speed, up to 0.85 and 1.29 Pa for the 134/56 and 134/89 dish respectively. When rotated at 100 rpm, the OSI was lowest, coinciding with the lowest τ_max_. Interestingly, the OSI peaked at 125 rpm for both dishes (Figure 5B). The OSI distribution for the 134/56 dish was similar for the 150- and 200-rpm situation. The OSI values of the 134/89 dish remained below 0.25, except for the 125-rpm situation. In terms of variations in shear stress direction the 134/89 on 19-mm orbit outperforms the 134/56 dish on 10-mm orbit. Since the OSI was not greatly affected by the increase from 150 to 200 rpm, 200 rpm is considered optimal as it can reach the highest shear levels.

Next to orbit size and orbital speed, the shear stress distribution is dependent on the amount of fluid in the dish. Here, we characterized the role of 2-, 3- and 4-mm initial fluid heights in both dishes at an orbital speed of 200 rpm. In general, more fluid led to a higher τ_max_ (Figure 6). In the 134/56 dish, the 2-mm situation led to a low τ_max_ of 0.34 Pa with little variation across the annular dish. Both 3- and 4-mm heights displayed higher shear stress values of 0.85 and 1.07 Pa, respectively. The τ_max_ distribution and OSI levels across the dish were similar for the 3- and 4-mm situation, whereas 2-mm fluid increased the OSI. The effect of fluid height was similar for the 134/89 dish on the 19-mm orbit. There was little difference between 3- and 4-mm of fluid, but the 2-mm case led to a shifted variation in τ_max_ and increased OSI values up to 0.69. The increase from 3- to 4-mm of fluid lowered the OSI distribution from a maximum of 0.25 to 0.10. Taken together, the 3-mm and 4-mm fluid heights are comparable for a 134/56 dish, whereas the 4-mm fluid height is the best candidate for 134/89, based on the OSI distribution.

### 3.3. Biological Characterization

To test the cellular shear stress response, endothelial cells were exposed to shear stress in the annular dish. Since our lab is equipped with a 10-mm orbital shaker, we used a 134/56 dish. We compared the cellular orientation in response to shear stress in the DiaD with a 35-mm and a 134-mm dish. Endothelial monolayers were subjected to shear stress on the orbital shaker at an orbital speed of 200 rpm and a medium height of 3 mm. Upon the onset of shear stress, the cells respond by modulating a wide variety of signaling pathways. In this work, we exposed the cells for 24 h to shear stress since after this period the shear stress response is reported to be stabilized [9]. The cells in the annular dish all aligned in the direction of the main shear direction (Appendix A). The cells in the center of the 35-mm and the 134-mm dish oriented randomly, whereas the cells on the edge aligned in the main shear direction (Figure 7A–C). 

We also analyzed the cellular response on a gene expression level in the 134/56 and 35-mm dish, as these were exposed to similar τ_max_, whereas the τ_max_ of the 134-mm dish was (45%) lower. The cells in the 35-mm dish were divided into three groups (pooled, edge and center) to separate the effect of different shear stress profiles for comparison with the 134/56 dish. Cells in the central region were obtained by scraping a circle 12 mm in diameter in the center [23]. The panel of shear stress responsive genes consisted of *KLF2*, *eNOS*, *MCP1* and *CDH2*. *KLF2* is a classical shear stress marker that is selectively upregulated under shear stress [12]. The expression of *eNOS* is higher in regimes with a low OSI, making it a suitable atheroprotective marker [13]. For atheroprone markers, we used *MCP1* and *CDH2*. Both genes were found to be upregulated in endothelial cells in atheroprone areas [8,9]. The gene expression *eNOS* and *KLF2* was increased for all cells that have been exposed to shear stress (Figure 7D). The expression in the pooled and edge groups of the 35-mm dish was slightly higher than the annular dish and the center groups. A similar reversed trend holds for the atheroprone markers *CDH2* and *MCP1*. The expression was lowest in the pooled and edge groups. The *CDH2* and *MCP1* expression was highest in the annular dish and the center of the 35-mm dish. Surprisingly, this indicated a more atheroprone shear in the 134/56 dish than in the 35-mm dish. Since the shear levels in these setups are similar, the atheroprone phenotype is likely related to the OSI levels which reach up to 0.3 in the 134/56 dish, whereas the 35-mm dish approaches 0 for a large region. A similar profile was seen for the 35 mm pooled dish vs. the edge fraction, which is likely due to the relatively small surface contribution of the region where the OSI is high. 

## 4. Discussion

Based on the combined computational and experimental data, we recommend the use of an orbital shaker with an orbit size of 19 mm in combination with a 134/89 dish at 200 rpm and 4 mm medium. These parameters give rise to an atheroprotective shear stress pattern throughout the dish, as shown by the low OSI. The disadvantage of this combination is that there is substantial τ_max_ variation. When low τ_max_ variation is desired, the 89/56 dish on the 10-mm orbit is a good alternative. In this dish, the maximum OSI is 0.11 and the variation in τ_max_ is lower, but the cell yield is lower. The cell culture area of a single 134/89 dish is 79 cm^2^, allowing for a typical cell count of ± 2 million cells, meeting the typical requirements of high input molecular techniques such as proteomics or chromatin conformation capture. These numbers can otherwise only be met by a cone-and-plate apparatus, but the throughput is limited to a single dish/plate. The annular dishes can be placed next to and stacked on top of each other, increasing the throughput significantly. Moreover, the orbital shaker is a more accessible platform than the cone-and-plate apparatus. 

To reach similar τ_max_ values as the 134/89 dish on the 19-mm orbit, one could increase the viscosity of the medium, since this directly scales with the shear stress level. The viscosity can be changed by supplementing the medium with polysaccharides such as methyl cellulose or xanthan gum [36]. These factors can increase the dynamic viscosity by four or five times, leading to higher shear stress levels. Alternatively, one could selectively seed the edge area of the 35-mm dish, preventing interference of atheroprone factors in the experiment. This could be done by scraping prior to the experiment or by selective passivation of the surface with agents that reduce cell adhesion such as Pluronic F-127 [37,38,39]. For the larger dishes, this approach is not possible, in these dishes the inner dish is needed to reach the atheroprotective shear stress conditions. 

The shear stress response is not solely dependent on the shear stress magnitude and direction. When studying endothelial cells under flow, the matrix and cell origin could also be considered. Here, we coated the dishes with collagen IV, since this is the main component of the basement membrane. Another commonly used coating is fibronectin, a protein that could be considered atheroprone as it is a protein produced during wound healing [40]. The source of the endothelial cells can play an even bigger role. Not only can the difference between arterial and venous origin play a role [41], there is also a wide heterogeneity between organs [42]. Moreover, it will be very interesting to use the annular dishes to investigate the role of different key signaling pathways in mechanosensing with both healthy and diseased cells. This can be in direct mechanotransduction regulators such as focal adhesion kinases or in cell–cell signaling pathways that are well-known as cell fate regulators [43]. An example of this is Notch signaling, an evolutionary conserved signaling pathway, that is emerging as an important regulator in the endothelial shear stress response [17,33,44,45]. 

In conclusion, this study characterized the shear stress distributions in different designs of orbital shaker setups. Although the DiaD is an accessible shear stress platform and a potential improvement over regular dishes on orbital shakers, it is important to calculate and validate the flow profiles. When carefully characterized, the DiaD can aid in improving our knowledge of vascular biology and expand our mechanistic knowledge of shear stress-induced response.

## Figures and Tables

**Figure 1 micromachines-11-00552-f001:**
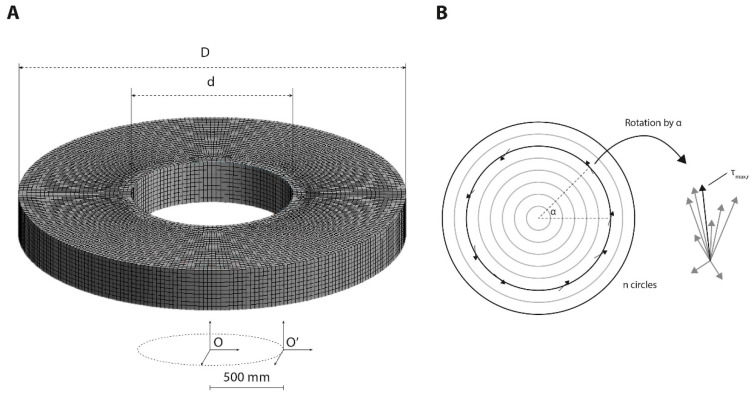
Mesh design and vector analysis. (**A**). Three-dimensional representation of the computational mesh used to model the different geometries, varying the inner (d) and outer (D) diameter. The coordinate system denotes the reference frame of the mesh. (**B**) Schematic representation of the calculation of the low oscillatory shear index (OSI) and τ_max_. The τ_max_ was extracted from each collection with similar a similar radial position. All vectors are first rotated to α=0 to obtain a time average to calculate the OSI for the variation in shear direction.

**Figure 2 micromachines-11-00552-f002:**
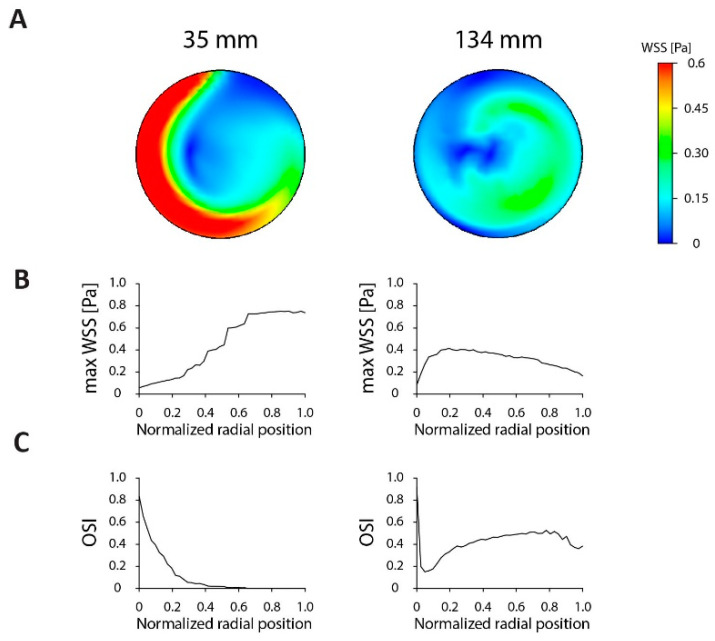
Comparison of shear stress distribution in standard culture dishes. (**A**) Heat map of the shear stress distribution in a six well plate (35 mm) and a 15-cm dish (134 mm). The dishes are scaled to equal size for shear stress comparison between dishes and the colored shear stress scale ranges from 0 to 0.6 Pa to display all shear stress distributions. For all situations, a 3-mm fluid height, 200-rpm rotational speed and a 10-mm orbit was used. (**B**) Maximum shear stress levels as a function of the radial position in the dish. (**C**) Oscillatory shear index (OSI) as a function of the radial position in the dish.

**Figure 3 micromachines-11-00552-f003:**
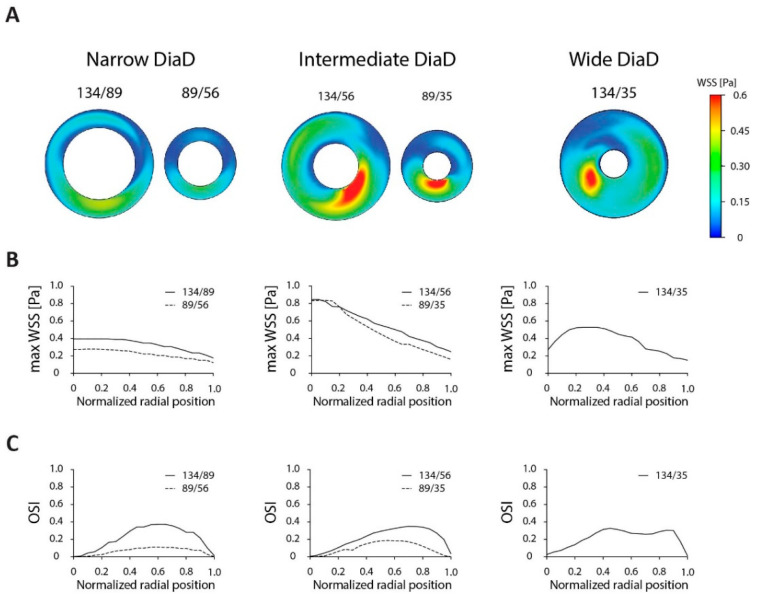
The effect of annular dish design on the distribution of wall shear stress. (**A**) Graphical representation of the shear stress distribution of different annular dish designs. The dish size is scaled to actual size. Dish dimensions are written as outer/inner dish diameter in millimeters. The graphical shear stress distribution scales to maximal 0.6 Pa to display the distributions in all designs. For the calculations, a 3-mm fluid height, 200-rpm rotational speed and a 10-mm orbit was used. (**B**) Maximum shear stress levels as a function of the radial position in the annular dish. (**C**) OSI as a function of the radial position in the annular dish.

**Figure 4 micromachines-11-00552-f004:**
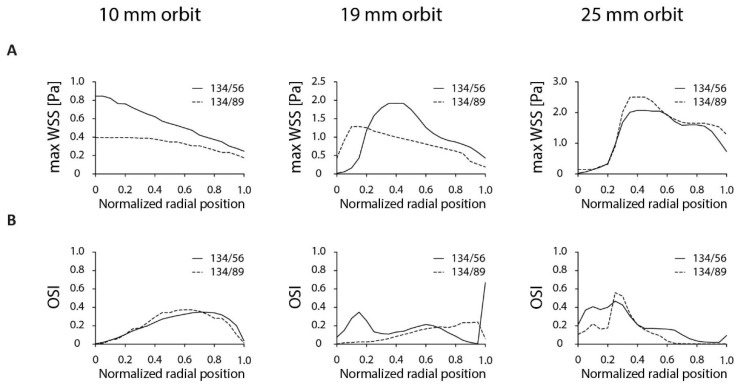
The effect of orbit size on shear stress distribution in the 134/56 and 134/89 design. (**A**) Maximum shear stress level at each radial position for the 134/56 and 134/89 annular dish on an orbital shaker with an orbit diameter of 10, 19 or 25 mm. For the calculations, a 3-mm fluid height and 200-rpm rotational speed was used. (**B**) OSI as a function of the radial position. Heat maps of the shear stress distributions can be found in Appendix A.

**Figure 5 micromachines-11-00552-f005:**
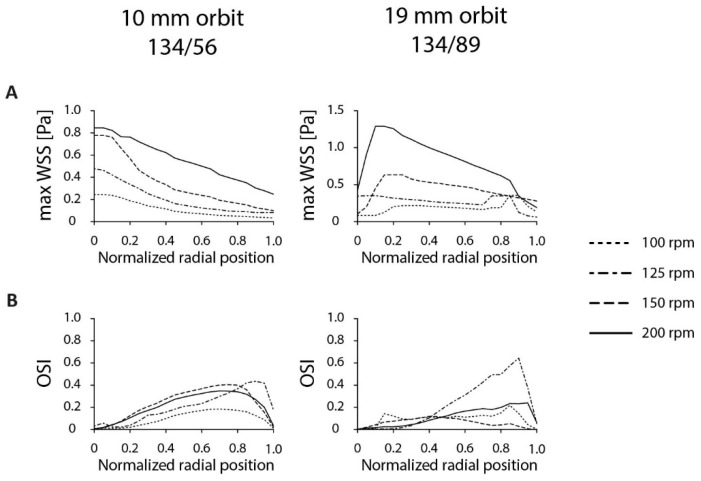
The influence of rotational speed on shear stress distribution. (**A**) Maximum shear stress level at each radial position for different orbital velocities in the 134/56 (10-mm orbit) and 134/89 (19-mm orbit) annular dish on an orbital shaker. For the calculations, a 3-mm fluid height was used. (**B**) OSI as a function of the radial position. Heat maps of the shear stress distributions can be found in Appendix A.

**Figure 6 micromachines-11-00552-f006:**
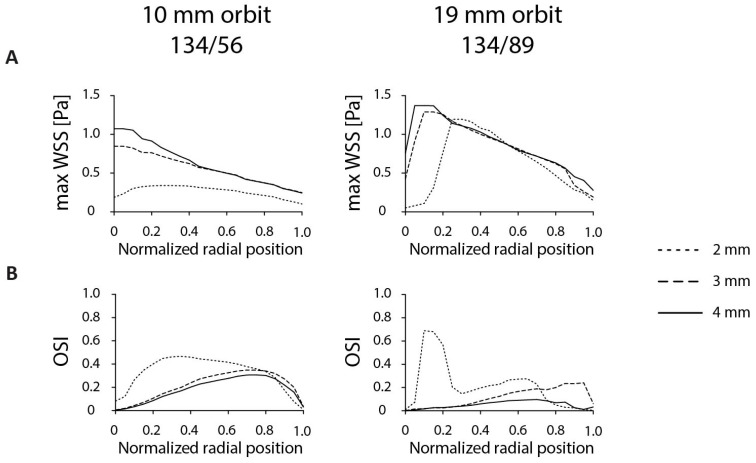
The influence of fluid levels in the annular dish on shear stress distribution. (**A**) Maximum shear stress level at each radial position with different starting fluid heights in the 134/56 (10-mm orbit) and 134/89 (19-mm orbit) annular dish on an orbital shaker. For the calculations, a 200-rpm orbital speed was used. (**B**) OSI as a function of the radial position. Heat maps of the shear stress distributions can be found in Appendix A.

**Figure 7 micromachines-11-00552-f007:**
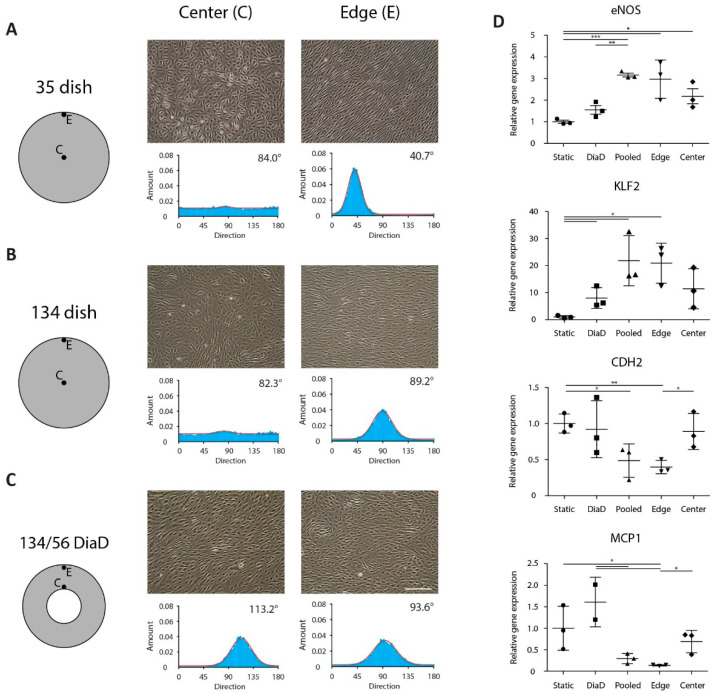
Endothelial cell response to shear stress experienced in the annular dish. (**A**–**C**) Widefield images of endothelial cells under shear stress at the center or the edge of a standard culture dish or an annular dish. The alignment directionality of the endothelial cells is quantified in the histograms below. Scalebar represents 200 μm. (**D**) Gene expression levels of the shear stress responsive genes *eNOS*, *KLF2*, *CDH2* and *MCP1* in endothelial cells under static conditions and in different (regions in) culture dishes. All genes demonstrate a response in gene expression in response to shear stress compared to the static controls. One measurement for the dish-in-a-dish (DiaD) group for *MCP1* was excluded based on Grubbs’ outlier test. Error bars are defined as the standard deviation. Significance is indicated at * *p* < 0.05, ** *p* < 0.01 and *** *p* < 0.001.

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
