# Peer review of "Computational Characterization of the Dish-In-A-Dish, A High Yield Culture Platform for Endothelial Shear Stress Studies on the Orbital Shaker"

_micromachines, 2020, doi:10.3390/mi11060552_

Round 1

Reviewer 1 Report

Rob et al proposes study on computational characterization and the effect of endothelial shear stress on high yield culture platform. Later part of the studies was focused on the validation by cell orientation and gene expression analysis. They conclude that, DiaD is an accessible and more improved version over regular dish to culture endothelial cells on orbital shakers. The computational approach was very well designed and brilliantly executed.

While interesting, the paper is a little more speculative and there may be some more direct ways drawing conclusions on the action of DiaD over the shear stress responsive genes the author selected to see the levels. The logical explanation over selection of these genes are not discussed properly in the result section (or as well as introduction). There should be a logical and experimental evidences (based on literatures) when there is a biased approach of selecting genes. Secondly, Endothelial cells require matrix or basement support for their growth (Invivo). Endothelial biologists will not buy the studies of endothelial cells in culture growing without any matrix in-vitro.  Basement matrix plays a vital role in the shear stress response of the endothelial cells. The authors have used Collagen IV matrix to culture HUVEC cells and for further studies. Focal adhesion kinases play a very important role for endothelial cells to matrix attachment, the aberrations to any of these mechanism leads to shear stress response (Song et al, 2002, PNAS).   Authors needs to discuss these aspects in terms of endothelial cells, basement membrane and their response in shear stress.  

References:

Li S, Butler P, Wang Y, Hu Y, Han DC, Usami S, Guan JL, Chien S. The role of the dynamics of focal adhesion kinase in the mechanotaxis of endothelial cells. Proceedings of the National Academy of Sciences. 2002 Mar 19;99(6):3546-51.

Author Response

Rob et al proposes study on computational characterization and the effect of endothelial shear stress on high yield culture platform. Later part of the studies was focused on the validation by cell orientation and gene expression analysis. They conclude that, DiaD is an accessible and more improved version over regular dish to culture endothelial cells on orbital shakers. The computational approach was very well designed and brilliantly executed.

We would like to thank reviewer 1 for his/her review, positive evaluation of the manuscript and valuable suggestions to improve its quality.

While interesting, the paper is a little more speculative and there may be some more direct ways drawing conclusions on the action of DiaD over the shear stress responsive genes the author selected to see the levels. The logical explanation over selection of these genes are not discussed properly in the result section (or as well as introduction). There should be a logical and experimental evidences (based on literatures) when there is a biased approach of selecting genes.

We agree that the selection of genes can be explained better. To clarify the selection of MCP1, CDH2, KLF2 and eNOS we have now introduced these genes in the introduction, accompanied with their relevant references. In the results/discussion section we explain the relevance of the selected genes. These changes are highlighted in red in line 39-40, 42-44 and 346-350.

Secondly, Endothelial cells require matrix or basement support for their growth (in vivo). Endothelial biologists will not buy the studies of endothelial cells in culture growing without any matrix in-vitro. Basement matrix plays a vital role in the shear stress response of the endothelial cells. The authors have used Collagen IV matrix to culture HUVEC cells and for further studies. Focal adhesion kinases play a very important role for endothelial cells to matrix attachment, the aberrations to any of these mechanism leads to shear stress response (Song et al, 2002, PNAS). Authors needs to discuss these aspects in terms of endothelial cells, basement membrane and their response in shear stress.

In our study we have used collagen IV as a coating since this is an important component of the basal lamina, the extracellular matrix (ECM) of the endothelial cells. We have added a section in the discussion about the role and influence of different ECM proteins different proteins that have been used as a coating in literature.  The response of endothelial cells is elaborated in the discussion as well. We further highlight the importance of disease modeling and give examples of both direct mechanotransduction mechanisms, such as FAK, and molecular signaling mechanisms that were until recently not known to be mechanosensitive or play a role in mechanotransduction, such as Notch signaling. These changes are highlighted in red in line 397-408.

The references that were added (ref 13, 22, 40-45) are also highlighted in red.

Reviewer 2 Report

With pleasure, I read your manuscript entitled: “Computational characterization of the dish-in-a-dish, a high yield culture platform for endothelial shear stress studies on the orbital shaker”. The authors present a numerical study of the shear stress generated by the DiaD in an orbital shaker under different conditions: design, speed, fluid height. The motivation of the work is well defined, with goals, methods, results and conclusions. The project brings interest to a very specific group of researchers (probably not so relevant to a wider academic community). The manuscript requires a few improvements to be considered for publication at Micromachines. Some points need better clarification. I have some remarks and questions:

  • Page 3, line 134: since the manuscript focus the numerical simulation, the mesh analysis studies should be included as supplementary material.
  • Page 4, section 2.2.1 – please include a photo of the fabricated PDMS DiaD system and the experimental setup.
  • Section 2.2.3 – why 24h to generate the shear stress? Was the effect of the shaking time studied?
  • The results section is very extensive and hard to read. I suggest the authors to divide section 3 in different subsections. More, in the manuscript some figures appear far from the region where they are referred (as figure 6 or 7). It makes the document hard to follow.
  • As the authors claim in the title and through the manuscript that the DiaD is a high yield platform, what is the maximum throughput supported by the DiaD platform and how does it compare to current solutions?
  • I would recommend a conclusions/future perspectives section after the results and discussion to enhance the main findings in the document and present the main challenges still to solve.
  • Did the authors perform any study comparing the effects (the response of the cells to the shear stress) between healthy and disease-like epithelial cells samples (as in cardiovascular disease or immune response – refs 3 and 4)? For instance, is it expected any difference in the cultured cells’ response to shear stress for healthy vs diseased cells, due, for instance, to their different mechanical properties? Please discuss this.

Author Response

With pleasure, I read your manuscript entitled: “Computational characterization of the dish-in-a-dish, a high yield culture platform for endothelial shear stress studies on the orbital shaker”. The authors present a numerical study of the shear stress generated by the DiaD in an orbital shaker under different conditions: design, speed, fluid height. The motivation of the work is well defined, with goals, methods, results and conclusions. The project brings interest to a very specific group of researchers (probably not so relevant to a wider academic community). The manuscript requires a few improvements to be considered for publication at Micromachines. Some points need better clarification. I have some remarks and questions:

 We would like to thank reviewer 2 for his/her review, positive evaluation of the manuscript and valuable suggestions to improve its quality.

Page 3, line 134: since the manuscript focus the numerical simulation, the mesh analysis studies should be included as supplementary material.

The mesh dependency was carried out on the largest structure. We used this structure since this is the most extreme case. We tested a mesh size of 1.0, 1.5, 2.0 and 2.5 mm. Based on the CPU time and output accuracy the 1.5 mm element size was chosen for meshing all structures. This information is now added to supplemental information 1. The reference to this supplemental information is highlighted in red in the methods section in line 143.

Page 4, section 2.2.1 – please include a photo of the fabricated PDMS DiaD system and the experimental setup.

To further clarify the DiaD design and experimental procedure an image of the fabricated DiaD with medium is added to the manuscript as supplemental figure 1. A reference to this figure was added in the methods section, highlighted in red in line 190.

Section 2.2.3 – why 24h to generate the shear stress? Was the effect of the shaking time studied?

In our work we apply a shear stress to the cells for 24 hours. Cells respond to shear stress in different ways at various time scales after the onset of fluid flow, as was clearly shown by detailed transcriptional analysis by Ajami and colleagues in 2017 (PMID 28973892). In the native situation, endothelial cells are constantly exposed to shear stress. To study their behavior in vivo we would like to avoid remodeling effect that are induced by the onset of fluid flow. Therefore, the endothelial cells are exposed to shear stress for 24 hours. This rationale was also added to the results section (highlighted in red in line 336-338) to motivate the 24 hour period.

The results section is very extensive and hard to read. I suggest the authors to divide section 3 in different subsections.

We agree that the results section is extensive. This section is now divided into three subsections: 3.1 Optimization of dish design, 3.2 Optimization of experimental parameters and 3.3 biological characterization. We also placed the last paragraphs, in which we discussed the results, in a separate discussion section. Again, these changes are highlighted in red in line 234, 272, 331 and 375.

More, in the manuscript some figures appear far from the region where they are referred (as figure 6 or 7). It makes the document hard to follow.

We agree that some figures are further apart from the references than others. This is mainly caused by the template of the journal which we have to follow. We moved some figures forward to improve to readability of the manuscript.

As the authors claim in the title and through the manuscript that the DiaD is a high yield platform, what is the maximum throughput supported by the DiaD platform and how does it compare to current solutions?

The DiaD is indeed a high yield platform. In our experience, a confluent layer of endothelial cells yields approximately 25.000 cells/cm2. This means the recommended 134/89 structure yields around 2 million cells. This order of magnitude can only be met with a cone-and-plate apparatus, but the throughput is lower. Theoretically, these numbers could be reached with the parallel plate setup, but that would require a large custom-made system. In the discussion we extended our comment on the cell yield with a comparison with current solutions, highlighted in red in line 384-387.

I would recommend a conclusions/future perspectives section after the results and discussion to enhance the main findings in the document and present the main challenges still to solve.

Did the authors perform any study comparing the effects (the response of the cells to the shear stress) between healthy and disease-like epithelial cells samples (as in cardiovascular disease or immune response – refs 3 and 4)? For instance, is it expected any difference in the cultured cells’ response to shear stress for healthy vs diseased cells, due, for instance, to their different mechanical properties? Please discuss this.

In our work we used healthy endothelial cells for biological characterization. The DiaD platform can indeed be a valuable tool to study molecular mechanisms in endothelial cells in diseases. At this point we did not include any disease models. We do believe that the cellular response is altered in diseased cells. For example, we recently demonstrated that absence of vimentin influences the Notch signaling under shear stress. We extended our discussion with speculation on the role of the matrix environment, cell source and molecular signaling under shear stress. Again, these changes are highlighted in red in line 397-408.

The references that were added (ref 13, 22, 40-45) are also highlighted in red.

Round 2

Reviewer 2 Report

The authors addressed the comments and concerns previously raised and improved the manuscript, making it suitable for publication.